# HARNESSING DISCRETE REPRESENTATIONS FOR CONTINUAL REINFORCEMENT LEARNING

## ABSTRACT

Reinforcement learning (RL) agents make decisions using nothing but observations from the environment, and consequently, heavily rely on the representations of those observations. Though some recent breakthroughs have used vector-based categorical representations of observations, often referred to as discrete representations, there is little work explicitly assessing the significance of such a choice. In this work, we provide a thorough empirical investigation of the advantages of representing observations as vectors of categorical values within the context of reinforcement learning. We perform evaluations on world-model learning, model-free RL, and ultimately continual RL problems, where the benefits best align with the needs of the problem setting. We find that, when compared to traditional continuous representations, world models learned over discrete representations accurately model more of the world with less capacity, and that agents trained with discrete representations learn better policies with less data. In the context of continual RL, these benefits translate into faster adapting agents. Additionally, our analysis suggests that the observed performance improvements can be attributed to the information contained within the latent vectors and potentially the encoding of the discrete representation itself.

## 1 INTRODUCTION

This work is motivated by the quest to design autonomous agents that can learn to achieve goals in their environments solely from their stream of experience. The field of reinforcement learning (RL) models this problem as an agent that takes actions based on observations of the environment in order to maximize a scalar reward. Given that observations are the agent's sole input when choosing an action (unless one counts the history of reward-influenced policy updates), the representation of observations plays an indisputably important role in RL.

The importance of observations becomes even more apparent when viewing proposed models of autonomous agents like the Common Model, identified by Sutton (2022), or JEPA, proposed by LeCun (2022). Nearly all of the components of these models, like the policy, value function, and world model, intake representations that originate from observations. Changes to observations are the most wide-reaching in the sense that they affect every part of the agent. Perhaps for this reason, both the Common Model and JEPA share a "perception" module that transforms observations into alternative representations before they are used by other components of the agent.

In this work, we examine the understudied yet highly effective technique of representing observations as vectors of categorical values, referred to in the literature as discrete representations (van den Oord et al., 2017; Hafner et al., 2021; Friede et al., 2023) — a method that stands in stark contrast to the conventional deep learning paradigm that operates on learning continuous representations. Despite the numerous uses of learned, discrete representations (Robine et al., 2021; Hafner et al., 2023; Micheli et al., 2023), the mechanisms by which they improve performance are not well understood. To our knowledge, the only direct comparison to continuous representations in RL comes from a single result from Hafner et al. (2021) in a subfigure in their paper. In this work, we dive deeper into the subject and investigate the effects of discrete representations in RL.

The successes of discrete representations in RL date back to at least as early as tile coding methods, which map observations to multiple one-hot vectors via a hand-engineered representation function

(Sutton & Barto, 2018, p. 217-222). Tile coding was popular prior to the proliferation of deep neural networks as a way to construct representations that generalize well, and has more recently been adopted to reduce interference between hidden units in neural networks (Ghiassian et al., 2020). Continuous alternatives exist — notably, radial basis functions (RBFs) could be viewed as a generalization of tile coding that produce values in the interval $[0, 1]$. Despite the superior representational capacity of RBFs, however, they have tended to under perform in complex environment with many input dimensions (An et al., 1991; Lane et al., 1992).

A similar comparison can be seen between the work of Mnih et al. (2015) and Liang et al. (2016). Mnih et al. train a deep neural network (DNN) to play Atari games, relying on the neural network to learn its own useful representation, or features, from pixels. In contrast, Liang et al. construct a function for producing binary feature vectors that represent the presence of various patterns of pixels, invariant to position and translation. From this representation, a linear function approximator is able to perform as well as a DNN trained from pixels.

Recent approaches to producing discrete representations have moved away from hand-engineering representations, and towards learning representations. Van den Oord et al. (2017), for example, propose the vector quantized variational autoencoder (VQ-VAE), a self-supervised method for learning discrete representations. VQ-VAEs perform comparably to their continuous counterparts, variational autoencoders (Kingma & Welling, 2014), and do so while representing observations at a fraction of the size. When applied to DeepMind Lab (Beattie et al., 2016), VQ-VAEs are able to learn representations that capture the salient features of observations, like the placement and structure of walls, with as little as 27 bits.

Similar representation learning techniques have also been successfully applied in the domain of RL. Hafner et al. (2021) train an agent on Atari games (Bellemare et al., 2013; Machado et al., 2018), testing both discrete and continuous representations. They find that agents learning from discrete representations achieve a higher average reward, and carry on the technique to a follow-up work (Hafner et al., 2023) where they find success in a wider variety of domains, including the Proprio Control Suite (Tassa et al., 2018), Crafter (Hafner, 2022), and Minecraft (Guss et al., 2019). Works like those from Robine et al. (2021) and Micheli et al. (2023) further build on these successes, using discrete representations to learn world models and policies. Work from Wang et al. (2022) find that representations that are more successful in transfer learning are often sparse and orthogonal, suggesting that these properties may underpin such successes of discrete representations.

The goal of this work is to better understand how discrete representations help RL agents. We use vanilla autoencoders (Ballard, 1987) to learn dense, continuous representations, FTA autoencoders (Pan et al., 2021) to learn sparse, continuous representations, and VQ-VAEs to learn fully discrete, binary representations. Inspired by the success of the Dreamer architecture (Hafner et al., 2021; 2023), we first examine how these different representations help in two distinct parts of a model-based agent: world-model learning and (model-free) policy learning. Observing that discrete and sparse representations specifically help when an agent's resources are limited with respect to the environment, we turn to the continual RL setting, where an agent must continually adapt in response to its constrained resources (Kumar et al., 2023). We particularly emphasize the benefits of discrete and sparse representations in continual RL, as the most large and complex environments are impossible to perfectly model and require continual adaptation to achieve the best performance possible (Sutton et al., 2007; 2022).

The primary contributions of our work include:

- Elucidating multiple ways in which discrete representations have likely played a key role in successful works in model-based RL.
- Demonstrating that the successes of discrete representations are likely attributable to the choice of one-hot encoding rather than the "discreteness" of the representations themselves.
- Identifying and demonstrating that discrete and sparse representations can help continual RL agents adapt faster.

## 2 BACKGROUND

This work primarily focuses on how to train agents that learn to achieve some goal by interacting with the environment. This problem is formulated as learning to select actions from states $S_t \in \mathcal{S}$,

that best maximize a given reward signal, $R_{t+1} \in \mathbb{R}$. We are specifically concerned with how to learn the parameters, $\boldsymbol{\theta}$, of a policy, $\pi_{\boldsymbol{\theta}}(A_t|S_t)$, that maps from states to a distribution over actions. The goal is to maximize the discounted return from the current state, which is given by $G_t \doteq \sum_{k=0}^{T} \gamma^k R_{t+k+1}$, where $T$ is the terminal time step, and $\gamma \in [0, 1]$ is the discount factor.

We use proximal policy optimization (PPO) (Schulman et al., 2017) to learn policies, which collects transitions through environment interaction, and then applies multiple epochs of stochastic gradient descent to weights that directly parameterize the policy. Training on the same data for multiple epochs results in a highly sample efficient algorithm. The sample efficiency of model-free RL algorithms like PPO can sometimes be further improved with the additional use of a world model (Sutton et al., 2008; Janner et al., 2019; Atkeson & Santamaría, 1997; Jin et al., 2018). Dyna (Sutton, 1991) is one such example of a framework for model-based RL that improves sample efficiency by learning from data generated by the model in a step called *planning*. In our work, we split model-based RL into its two components—world-model learning and (model-free) policy learning—and examine both components separately for a fine-grained view of how our solutions affect complex RL agents.

Both policy and world model architectures are split into two components in our work: a representation network (or encoder) that extracts a representation, and a problem-specific network that learns a policy or world model atop the learned representations. This decoupling can be beneficial in multiple ways (Lan et al., 2022; Barreto et al., 2017; Bellemare et al., 2019; Dabney et al., 2021), but we use it primarily as a means to carefully investigate how different representations affect learning. It allows us to swap out the encoder (both architecture and objective), while keeping the problem-specific model unchanged (aside from the input layer, which may vary in size).

With the exception of an end-to-end baseline, each of the encoders we use are trained with an observation reconstruction objective as part of a larger autoencoder model (Ballard, 1987). The autoencoder architecture compresses an observation into a bottleneck state before attempting to reconstruct it, forcing it to learn a representation that captures salient aspects of the observation. Each of the three types of learned representations we use in our work are produced by different autoencoder variants. We also evaluate the standard approach of end-to-end learning, where the representations are learned as a byproduct of the optimization process.

Dense, continuous representations are produced by a vanilla autoencoder.[1] Sparse, continuous representations also use a vanilla autoencoder, but the bottleneck layer outputs are passed through a Fuzzy Tiling Activation (FTA) (Pan et al., 2021). FTA is a function that produces sparse outputs by converting scalars to "fuzzy" one-hot vectors. The FTA representations act as a bridge between dense, continuous representations and discrete representations, and they have been established as a strong baseline known to yield strong results in RL (Miahi, 2022; Wang et al., 2022). Discrete representations are produced by a vector quantized-variational autoencoder (VQ-VAE) (van den Oord et al., 2017), which quantize the multiple outputs of the encoder to produce a vector of discrete values, also referred to as the *codebook*. The discrete representation we refer to in our work are comprised of multiple one-hot vectors, each representing a single, discrete value from the codebook. The details of these autoencoders are explained in more depth in Section A.1.

## 3 WORLD-MODEL LEARNING WITH DISCRETE REPRESENTATIONS

We begin our experiments by examining the benefits of using discrete representations when learning a sample-based world model.

### 3.1 ENVIRONMENTS

Throughout this work, we use the *empty*, *crossing*, and *door key* Minigrid environments (Chevalier-Boisvert et al., 2023), as displayed in Figure 1. In each environment, the agent receives pixel observations, and controls a red arrow that navigates through the map with `left`, `right`, and `forward` actions. The agent in the *door key* environment additionally has access to `pickup` and `use` actions to pickup the key and open the door. The *crossing* and *door key* environments are stochastic, with

---

[1]We also tested variational autoencoders (Kingma & Welling, 2014) in early model learning experiments, but were unable to find hyperparameters to made the method competitive. Future work may be able to improve upon this baseline with other variations like $\beta$-VAEs or VAEs with Gaussian mixture model priors.

each action having a 10% chance to enact a random, different action. The stochasticity increases the difficulty of learning a world model by increasing the effective number of transitions possible in the environments. The increase in difficulty widens the performance gap between different methods, which makes the results easier to interpret.

The environments are episodic, terminating when the the agent reaches the green square, or when the episode reaches a maximum length. The former yields a reward $R_t \in [0.1, 1]$ depending on the length of the episode (shorter episodes yield higher rewards), and the latter yields no reward. The reward is calculated with the standard Minigrid formula, $1 - 0.9\frac{t}{T}$, where $t$ is the current step and $T$ is the maximum episode length (dependent on the experiment). Though the environment is partially observable because the agent does not observe the current time step, this detail should not stop the agent from learning an optimal policy. Further environment details are displayed in Table 3 in the Appendix.



(a) Empty      (b) Crossing      (c) Door Key

Figure 1: Minigrid environments used in our experiments. We refer to these as the (a) *empty*, (b) *crossing*, and (c) *door key* environments. The agent receives lower-resolution RGB arrays representing pixels as observations.

## 3.2 LEARNING WORLD MODELS

We train autoencoders and world models on a static dataset, $\mathcal{D}$, of one million transition tuples, $(s, a, s')$, collected with random walks. In each episode, the environment terminates when the agent reaches the green square or after 10,000 steps. Training occurs in two phases: first the autoencoder is trained, and then a transition model is trained over the fixed representations.

Observations are 3-dimensional RGB arrays, so we use convolutional and deconvolutional neural networks (LeCun et al., 1989) for the encoder and decoder architectures. The encoder architecture is similar to the IMPALA network (Espeholt et al., 2018), but the size of the bottleneck layer is chosen with a hyperparameter sweep. Architectural details are given in Section A.3. All of the autoencoders are trained with a mean square error reconstruction loss, and the VQ-VAE with additional loss terms as detailed in Section A.1. Training for both autoencoders and world models use the Adam optimizer (Kingma & Ba, 2015) with hyperparameter values of $\beta_1 = 0.9$, $\beta_2 = 0.999$, and a step size of $2 \times 10^{-4}$. Training continues for a fixed number of epochs, until near-convergence, at which point the model weights are frozen and world model learning begins.

World models learned over latent representations take a latent state, $\mathbf{z}$, and an action, $a$, as input to predict the next latent state, $\hat{\mathbf{z}}' = w_\psi(\mathbf{z}, a)$, with an MLP, $w_\psi$. World models learned over continuous representations, or *continuous world models*, consist of three layers of 64 hidden units (32 in the *crossing* environment), and rectified linear units (ReLUs) (Agarap, 2018) for activations. In *discrete world models*, the MLP is preceded by an embedding layer that converts discrete values into a continuous, 64-dimensional vectors. The loss for both world models is given by the difference between the predicted next latent state and the ground-truth next latent state. The continuous world model outputs a continuous vector and uses the squared error loss. The discrete model outputs outputs multiple vectors of categorical logits and uses a categorical cross-entropy loss over each. [2] All world models are trained with 4 steps of hallucinated replay as described by Talvitie (2017), which entails feeding outputs of the model back in as new inputs. Figures 10 and 11 in the Appendix depict this training process for continuous and discrete world models.

Our aim is to train sample models — models that emulate the environment by producing outcomes with frequency equivalent to that of the real environment. This is more difficult in stochastic environments because our current training procedure would result in expectations models, where predictions are weighted averages over possible outcomes. To instead learn sample models, we augment our models using the method proposed by Antonoglou et al. (2022). This approach learns a distribution over potential outcomes, and samples from them when using the world model. We provided a more detailed explanation and relevant hyperparameters in Section A.2.

---

[2] We also experimented with a squared error loss for the discrete world model and found it made little difference in the final world model accuracy.

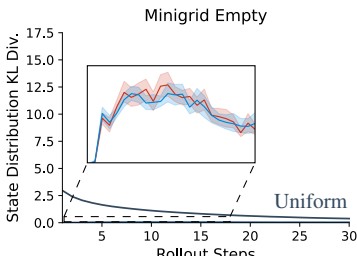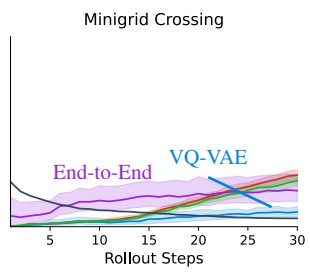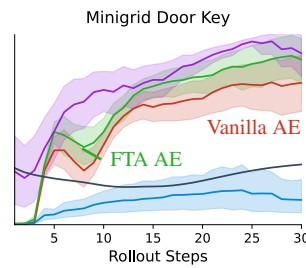

Figure 2: The KL divergence between the ground-truth state distribution and the world model induced state distribution. Lower values are better, indicating a closer imitation of the real environment dynamics. The VQ-VAE and Vanilla AE learn near-perfect models in the *empty* environment, so the curves are so close to zero that they are not visible without maginification. FTA AE and End-to-End experiments were not run in the *empty* environment because of the triviality. Curves depict averages over 20 runs with 95% confidence intervals.

## 3.3 EXPERIMENTS

The goal of this first set of experiments is to measure **how the representation of the latent space affects the ability to learn an accurate world model**. Unfortunately, this is not as simple as comparing a predicted latent state to the ground-truth latent state, as multiple outcomes may be possible for any given state-action pair. To account for this, we look at distributions over many transitions instead of the outcomes of single transitions. Specifically, we measure the differences between state distributions induced by a chosen behavior policy in the real environment and the same policy in an environment simulated by the learned transition model. Accurate world models should produce state distributions similar to that of the real environment, and inaccurate models should produce state distributions that differ. Figure 12 in the Appendix contains a visualization that helps build an intuition of how state distributions may differ, which we will discuss in more detail later.

The ability of world models to simulate trajectories outside of their training data is one of their major benefits, so to reflect this use case, we chose behavior policies that differ from the data collection policy. We use a random policy for the *empty* environment, a policy that explores the right half of the grid in the *crossing* environment, and a policy that navigates directly to the goal in the *door key* environment. Each of the policies are used to simulate 10,000 episodes in the real environment, and 10,000 episodes where the transition dynamics are simulated entirely by the learned world model. Episodes are cut off early, or frozen at the terminal state to reach exact 30 steps of interaction. We then compare the difference between state distributions at each step by measuring the KL divergence between the induced and ground-truth state distributions. A lower KL divergence is better, indicating that a model predicts outcomes more similar to the real environment.

We include two baselines in our comparisons that do not include auxiliary autoencoder objectives: the uniform baseline and the end-to-end baseline. The uniform baseline predicts a uniform distribution over all states and is strong when the agent's target policy leads it to spread out, like in a random walk. The end-to-end baseline shares an architecture equivalent to the vanilla autoencoder, but the full model is trained end-to-end with a next-observation reconstruction loss, and the size of the latent state is re-tuned in a separate hyperparameter sweep. This is the standard setup in deep RL.

### 3.3.1 MODEL ROLLOUTS

We roll out the trained world models for 30 steps and evaluate their accuracy, plotting the results in Figure 2. Although all of the methods perform the same in the *empty* environment, the gap in accuracy widens as the complexity progressively increases in the *crossing*, and then in the *door key* environment.

We examine visualizations of trajectories to better understand the patterns observed in Figure 2, showing two visualizations that most clearly represent these patterns in Figures 12 and 13 in the Appendix. The trajectories predicted by the continuous models (Vanilla AE and FTA AE) in the *crossing* environment rarely make it across the gap in the wall, which manifests as a steady increase in the

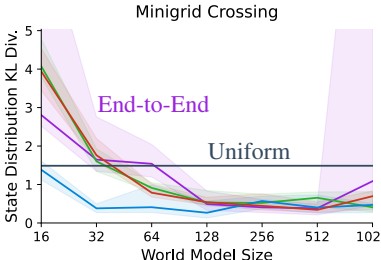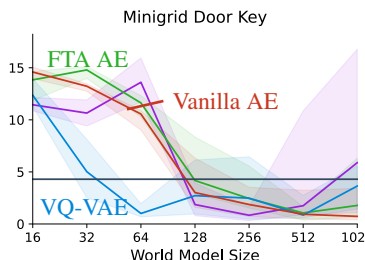

Figure 3: The KL divergence between the ground-truth and world model induced state distributions, averaged over 30 steps. Lower is better, indicating a closer imitation of the real environment dynamics. The x-axis gives the number of hidden units per layer for all three layers of the world model. Each point depicts the median over 20 runs with 95% confidence intervals. Error bars are high for the end-to-end method likely due to a few divergent runs. Training the end-to-end model is harder because gradients for multiple objectives must be passed back in time through multiple steps.

KL divergence starting around step 14. The performance of the continuous model in the *door key* environment suffers much earlier as the model struggles to predict the agent picking up the key, and again as the model struggles to predict the agent passing through the door. Notably, these two actions occur infrequently in the training data because the training data is generated with random walks, and because they can only happen once per episode even when they do occur. *Stated concisely, the discrete world model more accurately predicts transitions that occur less frequently in the training data.*

### 3.3.2 SCALING THE WORLD MODEL

Despite sweeping over the latent vector dimensions of the vanilla and FTA autoencoders in the hyperparameter sweep, we were unable to find an encoder architecture that enabled either of the continuous world models to adequately learn transitions underrepresented in the training data. **Either the discrete representations allow learning something that is not learnable with the continuous representations, or the fixed size of the world model is limiting the continuous model's performance.** We test the latter hypothesis by varying the size of the world model while tuning the latent dimensions of each autoencoder as described in Section A.3. We plot the average performance of each world model in Figure 3.

In the plot, an interesting pattern emerges: the performance of all methods become indistinguishable beyond a certain size of the world model. Only when the environment dynamics cannot be modeled near-perfectly, due to the limited capacity of the world model, do the discrete representations prove beneficial. As the size of the world model shrinks, the performance of the continuous models degrade more rapidly. This observation aligns with the findings in the previous section, where the performance gap between models widened with the complexity of the environment. Both results converge to the same conclusion: *the VQ-VAE discrete representations enable learning more of the world with less capacity, relative to the size of the environment*. This gap is notable especially when the world is much larger than what the agent has capacity to model. In this setting, discrete representations are arguably favorable because they allow an agent to learn more despite its limited capacity.

### 3.3.3 REPRESENTATION MATTERS

Our experiments demonstrate the potential advantage of using VQ-VAE latents, but latent spaces are defined both by the information they represent—informational content—and by the way that information is structured—representation. Our goal in the previous experiments was to measure how representation alone affects performance, but we do not directly control for information content—i.e. the different bottleneck structures of a vanilla AE and a VQ-VAE may change what is learned. Our next experiment controls for this factor as we ask the question: **do the benefits of discrete world models stem from the representation or from the informational content of the latent states?**

To answer this question, we rerun the model learning experiment with two types of latents, both produced by the same VQ-VAE but represented in different ways. Generally, the outputs of a VQ-VAE encoder are quantized by "snapping" each latent to the nearest of a finite set of embedding vectors. The resulting *quantized latents* are discrete in the sense that each can take only a finite number of

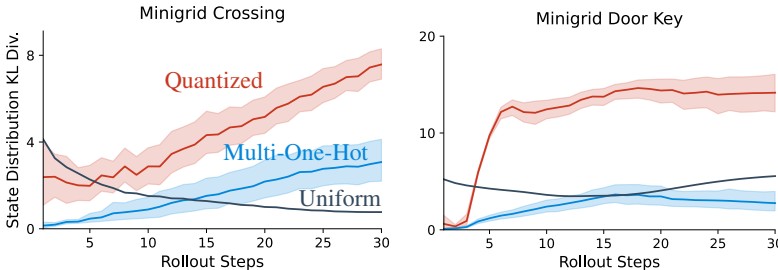

Figure 4: The KL divergence between the ground truth state distribution and the world model induced state distribution. Lower values are better, indicating a closer imitation of the real environment dynamics. Both methods use the same VQ-VAE architecture, but represent the information in different ways. Curves depict averages over 20 runs with 95% confidence intervals.

distinct values, but are element-wise continuous. In our work, we alternatively represent latents as (one-hot encoded) indices of the nearest embedding vectors, which are element-wise binary. Both of these methods encode the same informational content and can produce latents of the same shape, but have different representations. If the representation of the latent space does not matter, then we would expect models learned over both representations to perform similarly.

We prepare the experiment by constructing architecturally equivalent world models with quantized and multi-one-hot representations. The number and dimensionality of the embedding vectors are set to 64 so that both representations take the same shape. The quantized model is trained with the squared error loss, but otherwise both models follow the same training procedure.

We plot the accuracy of both models in Figure 4, where we see multi-one-hot representations vastly outperform quantized representations despite both being discrete and semantically equivalent. These results support the claim that *the representation, rather than the informational content, is responsible for the superior performance of the VQ-VAE latents* in our experiments. Our results also suggest that *the superior performance of discrete representations is not necessarily attributable to their "discreetness", but rather to their sparse, binary nature*. Both quantized and multi-one-hot representations are discrete and semantically equivalent, yet yield different results. These results reveal that the implicit choice of representing discrete values as multi-one-hot vectors is essential to the success of discrete representations, yet to our knowledge, such a choice is not discussed in any prior work.

## 4 MODEL-FREE RL WITH DISCRETE REPRESENTATIONS

As we progress to the full reinforcement learning problem, we face new challenges, like that of learning from non-stationary distributions. Our first experiments of this section aim to understand the effects of using discrete representations in the standard, episodic RL setting. After identifying a clear benefit, we progress to the continual RL setting with continually changing environments Abbas et al. (2023) as a proxy for environments that are too big for the agent to perfectly model.

We train all RL agents in this section with the clipping version of proximal policy optimization (PPO) (Schulman et al., 2017). Instead of observations, the policy and value functions intake learned representations. Separate networks are used for the policy and value functions, but both share the same architecture, an MLP with two hidden layers of 256 units and ReLU activations. We sweep over select hyperparameters for PPO and over autoencoder hyperparameters as described in Section 2.

Training loops between collecting data, training the actor-critic model, and training the autoencoder, and is detailed in Algorithm 2 in the Appendix. This setup differs from previous experiments in that environment interaction and the training of each component happen in tandem instead of in separate phases. The objectives, however, remain separate; PPO gradients only affect the policy and value function weights, while autoencoder gradients only affect the encoder. Only the end-to-end baseline is an exception, in which the entire model is trained with PPO, as is often standard in deep RL.

Agents are trained in the *crossing* and *door key* environments shown in Figure 1. The maximum episode length is set to 400 in the *crossing* environment and 1,000 in the *door key* environment.

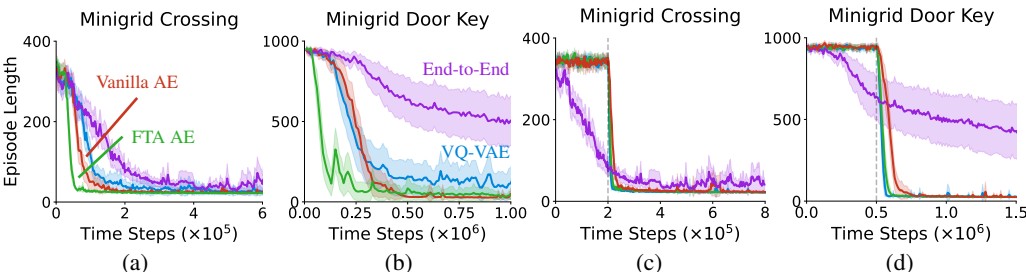

Figure 5: Performance of RL agents as measured by episode length with a 95% confidence interval over 30 runs. Lower is better. (a-b) Agents are trained with PPO and autoencoder objectives from the beginning. (c-d) The PPO objective is introduced only after the dotted line (with the exception of the end-to-end method).

## 4.1 EPISODIC RL

We train RL agents with each type of representation in the *crossing* and *door key* environments, plotting the results in Figures 5a and 5b. All of the methods with an explicit representation learning objective perform better than end-to-end RL. In a reverse from the previous model learning results, the VQ-VAE now performs the worst of all the representation learning methods. Inspecting the autoencoder learning curves in Figure 15 in the Appendix, however, reveals an important detail: all of the autoencoders learn at different speeds. If the speed of the RL learning updates is our primary concern (whether it actually is will be discussed later), then the learning speed of the autoencoder is a confounding factor. We address this by delaying PPO updates until all autoencoders are trained to around the same loss and plot the results in Figures 5c and 5d. Though the gap in performance in the new results looks small, *the VQ-VAE and FTA autoencoder methods converge with around two to three times less PPO updates than the vanilla autoencoder*.

## 4.2 CONTINUAL RL

While static Minigrid environments can test these representation learning methods to an extent, they do not reflect the vastness of the real world. When the size of the world and the complexity of its problems dwarf that of the agent, the agent will lose its ability to perfectly model the world and learn perfect solutions (Sutton et al., 2022). The agent must instead continually adapt in response to its limited capacity if it is to best achieve its goal(s) in this continual RL setting (Kumar et al., 2023). Given the ability of these representation learning methods to expedite policy learning, they may be well suited for the continual RL setting, where fast adaptation is key.

To test this hypothesis, we modify the previous experimental RL setup by randomizing the layout of the *crossing* environment every 40,000 steps, and the layout of the *door key* environment every 100,000 steps, as is similarly done in related work (Taylor & Stone, 2009; Khetarpal et al., 2022; Abbas et al., 2023). All of the same items and walls remain, but their positions are randomized, only the positions of the goal and outer walls staying constant. Example layouts are shown in Figure 14 in the Appendix. By only changing the environment after a long delay, we create specific points in the learning process where we can observe the difference between how the different types of representation methods adapt to change. The RL training process otherwise stays the same, and is specified in Algorithm 2 in the Appendix. With only this modification to the environments, we rerun the previous RL experiment with a delayed PPO start, and plot the results in Figures 6a and 6b.

We observe a spike in the episode length each time the environment changes, indicating that the agents' previous policies are no longer sufficient to solve the new environments. While the representation learning methods clearly outperform end-to-end training, the confidence intervals overlap at many time steps. If we instead, however, consider the average reward accumulated by each method per layout as displayed in Table 4 in the Appendix, a clear ranking emerges. In the *crossing* environment we see VQ-VAE > FTA AE > Vanilla AE, and in the *door key* environment we see VQ-VAE > FTA AE ≈ Vanilla AE.

While the slower initial learning speed of the VQ-VAE hinders its ability to maximize reward at the beginning of the training process (when PPO updates are not delayed), it does not seem to hinder its

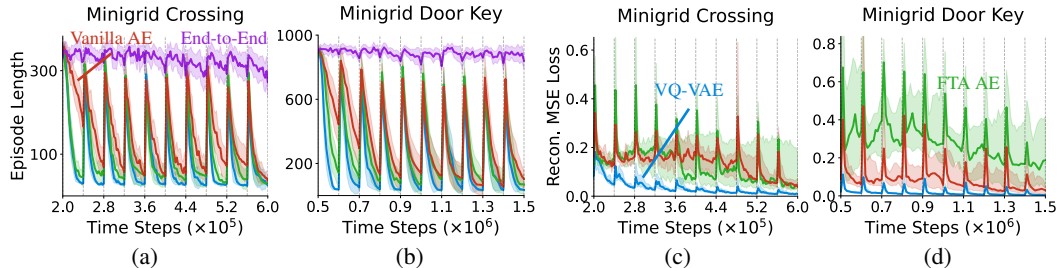

Figure 6: (a-b) Mean agent performance as the environments change at intervals indicated by the dotted, black lines. Lower is better. (c-d) Median encoder reconstruction loss. Lower peaks mean the representation generalizes better, and a quicker decrease means the autoencoder is learning faster. Overall, a lower reconstruction loss is better. (a-d) Results are averaged over 30 runs and depict 95% confidence intervals. Performance is plotted after an initial delay to learn representations, after which all methods are trained with PPO. Refer to Figure 16 for the full figure.

ability to adapt after an initial representation has already been learned. Inspecting the reconstruction loss of both autoencoders, plotted in Figures 6c and 6d, shows that the VQ-VAE's reconstruction loss increases much less when the environment changes. The shorter spikes suggest that the VQ-VAE representations generalize better, allowing them to adapt faster when the environment changes.

With these results, we return to the prior question: can multi-one-hot representations be beneficial in RL even if the initial representation is learned slower? We argue in the affirmative. If we consider continually learning RL agents in the big world setting, where the goal of the agent is to maximize reward over its lifetime by quickly adapting to unpredictable scenarios, then the cost of learning an initial representation is easily amortized by a lifetime of faster adaptation.

## 5 CONCLUSION & FUTURE WORK

In this work, we explored the effects of learning from discrete and sparse representations in three modules that are commonly found in models of intelligent agents: a world model, a value function, and a policy. When learning a world model, discrete, multi-one-hot representations enabled accurately modeling more of the world with less resources. When in the model-free RL setting, agents with multi-one-hot or sparse representations learned to navigate to the goal and adapt to changes in the environment faster.

Our study underscores the advantages of multi-one-hot representations in RL but leaves several questions of deeper understanding and extrapolation to future work. We show that one-hot encoding is crucial to the success of discrete representations, but do not disentangle multi-one-hot representations from purely binary or sparse representations in our experiments. While the results of the FTA autoencoder and prior work (Wang et al., 2022) suggest that sparsity and orthogonality are major factors in the success of multi-one-hot representations, the evidence is not conclusive. Future work could also experiment with different methods of producing discrete representations or apply these methods to a wider variety of environments, beyond the inherently discrete domain of Minigrid. Prior work on DreamerV3 (Hafner et al., 2023) and the success of VQ-VAEs in the domain of computer vision (van den Oord et al., 2017; Nash et al., 2021; Esser et al., 2021; Hong et al., 2022) already suggest that this method will extrapolate and scale to larger environments.

Regardless of these open questions, our results implicate multi-one-hot representations learned by VQ-VAEs as a promising candidate for the representation of observations in continual RL agents. If we care about agents working in worlds much larger than themselves, we must accept that they will be incapable of perfectly modeling the world. The agent will see the world as forever changing due to its limited capacity, which is the case in complex environments like the real world (Sutton et al., 2022; Kumar et al., 2023). If we wish to address this issue in the representation learning space, agents must learn representations that enable quick adaptation, and are themselves quick to adapt (Sutton et al., 2007). Multi-one-hot representations learned by VQ-VAEs do exactly that, and provide a path towards ever more efficient, continually learning RL agents.

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

# A   APPENDIX

## A.1   AUTOENCODERS EXPLAINED

In this work, we opt to learn representations with autoencoders, neural networks with the objective of reconstructing their own inputs. Autoencoders can be decomposed into an encoder, $f_\theta$, that projects the input into a latent space, and a decoder, $g_\phi$, that attempts to reverse the transformation. Where $\mathbf{x} \in \mathbb{R}^n$ is an observation input to the encoder, the corresponding latent state is given by $\mathbf{z} = f_\theta(\mathbf{x}) \in \mathbb{R}^k$, and the goal is to learn parameters $\theta$ and $\phi$ such that $g_\phi(f_\theta(\mathbf{x})) = \mathbf{x}$. We achieve this by minimizing the squared error between the input and the reconstruction over observations sampled from some dataset, $\mathcal{D}$:

$$\mathcal{L}_{\text{ae}} = \mathbb{E}_{\mathbf{x} \sim \mathcal{D}}\Big[||\mathbf{x} - g_\phi(f_\theta(\mathbf{x}))||_2^2\Big]. \tag{1}$$

Because the latent space of an autoencoder is constrained (generally by size, and sometimes by regularization), the model is encouraged to learn properties of the input distribution that are the most useful for reconstruction. We refer to this type of autoencoder, where the latent states are represented by vectors of real-valued numbers, as a vanilla autoencoder. An overview of the model is depicted in Figure 7.

To learn discrete representations, we use an autoencoder variant called a vector quantized variational autoencoder (VQ-VAE) van den Oord et al. (2017). VQ-VAEs also use an encoder, a decoder, and have the same objective of reconstructing the input, but include an additional *quantization* step that is applied to the latent state between the encoder and decoder layers. After passing the input through the encoder, the resultant latent state $\mathbf{z}$ is split into $k$ latent vectors of dimension $d$: $\{\mathbf{z}_1, \mathbf{z}_2, \ldots, \mathbf{z}_k\} \in \mathbb{R}^d$. Each latent vector is quantized, or "snapped", to one of $l$ possible values specified by a set of embedding vectors. The quantization function uses $l$ embedding vectors of dimension $d$, $\{\mathbf{e}_1, \mathbf{e}_2, \ldots, \mathbf{e}_l\} \in \mathbb{R}^d$, which are learned parameters of the VQ-VAE.

The quantization happens in two phases. First, each latent vector is compared to every embedding vector using the L2 norm, and indices of the most similar embedding vectors are returned:

$$c_i = \arg\min_j \|\mathbf{z}_i - \mathbf{e}_j\|_2, \text{ for all } i = 1, 2, ..., k. \tag{2}$$

The resultant vector of integers $\mathbf{c}$ is called the *codebook*, and indicates which embedding vectors are the most similar to each latent vector. In the second phase, the indices in the codebook are used to retrieve their corresponding embeddings, producing the quantized latent vectors:

$$\mathbf{z}'_i = \mathbf{e}_{c_i}, \text{ for all } i = 1, 2, ..., k. \tag{3}$$

The quantized vectors $\{\mathbf{z}'_1, \mathbf{z}'_2, \ldots, \mathbf{z}'_k\} \in \mathbb{R}^d$ are the final output of the quantization function, and are concatenated before being passed to the decoder. The full architecture is depicted in Figure 8.

Because the quantization process is not differentiable, a *commitment loss* is added to pulls pairs of latent states and their matching embeddings towards each other. If latent vectors are always near an existing embedding, then there will be minimal difference between all $\mathbf{z}_i$ and $\mathbf{z}'_i$, and we can use the straight-through gradients trick Bengio et al. (2013) to pass gradients directly back from $\mathbf{z}'$ to $\mathbf{z}$ with no changes. Combining the reconstruction and commitment losses, the full objective is given by the minimization of

$$\mathcal{L}_{\text{vqvae}} = \mathbb{E}_{\mathbf{x} \sim \mathcal{D}}\left[||\mathbf{x} - g_\phi(q_{\mathbf{e}}(f_\theta(\mathbf{x})))||_2^2 + \beta \sum_{i=1}^{k} \|\mathbf{z}_i - \mathbf{e}_{\mathbf{z}_i}\|_2^2\right], \tag{4}$$

where $q_{\mathbf{e}}$ is the quantization function, $\beta$ is a hyperparameter that weights the commitment loss, and $\mathbf{e}_{\mathbf{z}_i}$ is the closest embedding vector to $\mathbf{z}_i$. In practice, the speed at which the encoder weights and embedding vectors change are modified separately by weighting the gradients of both modules individually. We use a value of $\beta = 1$ in our work, and scale the embedding updates with a weight of 0.25.

The discrete representations we use for downstream tasks RL tasks are different from the quantized vectors that are passed to the decoder. We instead use one-hot encodings of the values in the codebook:

$$o_{ij} = \begin{cases} 1 & \text{if } j = c_i, \\ 0 & \text{otherwise} \end{cases} \quad \text{for } j = 1, 2, \ldots, l. \tag{5}$$

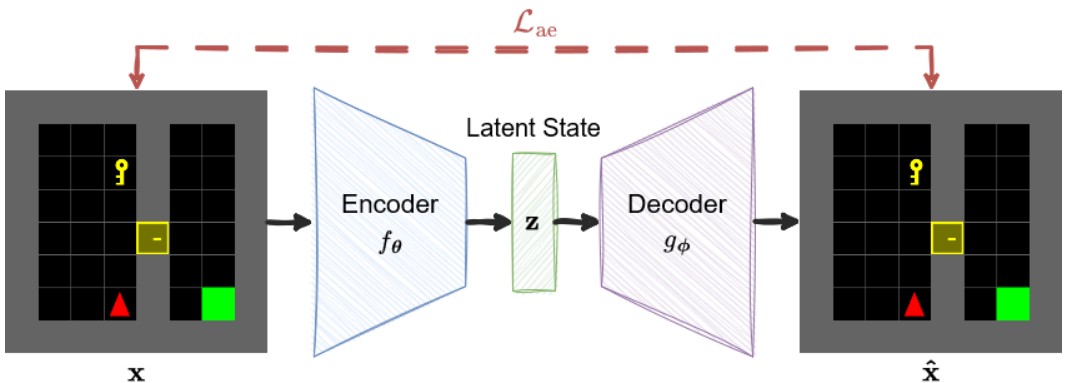

Figure 7: Depiction of a vanilla autoencoder with a continuous latent space. The input $\mathbf{x}$ is encoded with $f_\theta$ to produce a latent state $\mathbf{z}$, which is decoded by $g_\phi$ to produce the reconstruction $\hat{\mathbf{x}}$. The model is trained to minimize the distance between the input and reconstruction with the reconstruction loss $\mathcal{L}_{\mathrm{ae}}$.

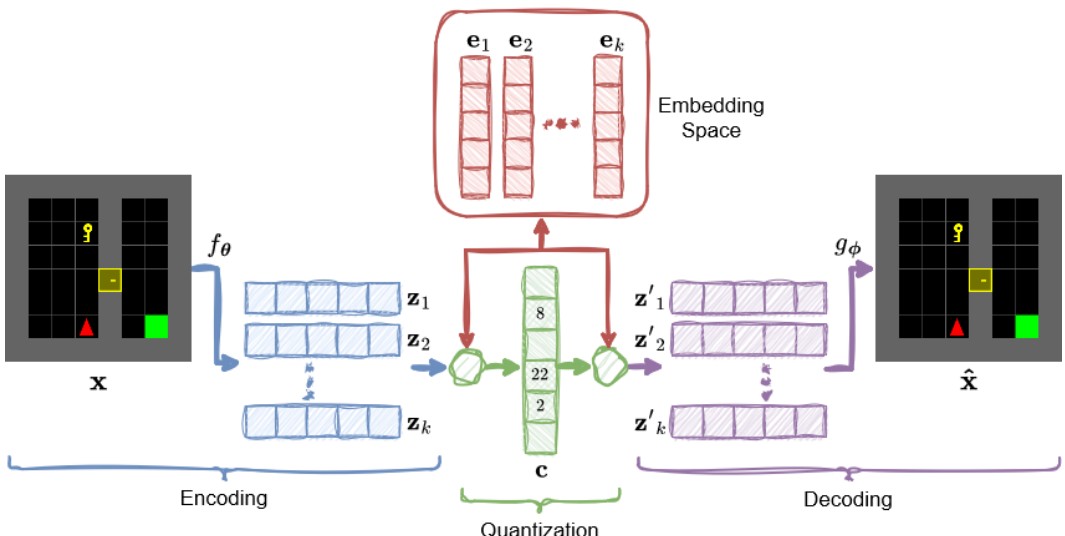

Figure 8: Depiction of the VQ-VAE architecture. The input $\mathbf{x}$ is encoded with encoder $f_\theta$ to produce latent vectors $\{\mathbf{z}_1, \mathbf{z}_2, \ldots, \mathbf{z}_k\} \in \mathbb{R}^d$. In the first green circle, each latent vector is compared to every embedding vector to produce codebook $\mathbf{c}$, a vector of indices indicating the most similar embedding vectors (example values are depicted). In the second green circle, the indices are transformed into their corresponding embedding vectors to produce quantized vectors $\{\mathbf{z}'_1, \mathbf{z}'_2, \ldots, \mathbf{z}'_k\} \in \mathbb{R}^d$. The quantized vectors are then decoded by $g_\phi$ to produce the reconstruction $\hat{\mathbf{x}}$. Our work uses one-hot encodings of the codebook $\mathbf{c}$ as discrete representations.

The result is a series of one-hot vectors $\{\mathbf{o}_1, \mathbf{o}_2, \ldots, \mathbf{o}_k\} \in \mathbb{R}^l$ that represent a single state, which we refer to as a multi-one-hot encoding or discrete representation.

## A.2 STOCHASTIC WORLD MODELS

We use a variant of the method proposed by Antonoglou et al. (2022) to learn sample models for stochastic environments. The method works similarly to a distribution model, first learning a distribution over possible outcomes during training, and then sampling from that distribution during evaluation. The problem faced by most distribution models is how to represent a distribution over a complex state space (or latent space in our case). Antonoglou et al. circumvent this problem by

learning an encoder $e$ that discretizes each state-action pair, mapping it to a single, $k$-dimensional one-hot vector we call the outcome vector. Each of the possible $k$ values represents a different outcome of the transition.

The high-level idea is that while directly learning a distribution over full latent states is intractable, learning a categorical distribution over a limited, discrete set of outcomes (the outcome distribution) is possible. Whenever we wish to use the world model, we can sample from the outcome distribution and include the one-hot outcome vector as an additional input to the world model, indicating which of the $k$ outcomes it should produce. Table 1 in provides the relevant hyperparameters for this method.

Table 1: Stochastic sample model hyperparameters

| Hyperparameter | Value |
| --- | --- |
| Bin count | 32 |
| Discretization projection | 256, 256 |
| Prediction projection | 256, 256 |

## A.3 AUTOENCODER ARCHITECTURE

The vanilla autoencoder, FTA autoencoder, and VQ-VAE use the same encoder and decoder architecture, only differing in the layer that produces the latent state. The decoder is a mirror of the encoder, reversing each of the shape transformation, so we describe only the encoder architecture. The encoder starts with three convolutional layers with square filters of sizes $\{8, 6, 4\}$, channel of sizes $\{64, 128, 64\}$, strides of $\{2, 2, 2\}$ (or $\{2, 1, 2\}$ for the crossing environment), and uniform padding of $\{1, 0, 0\}$. Each convolutional layer is followed by a ReLU activation. The downscaling convolutions are followed by an adaptive pooling layer that transforms features into a shape of $(k \times k \times 64)$, and finally a residual block (He et al., 2016) consisting of a convolutional layer, batch norm (Ioffe & Szegedy, 2015), ReLU, convolutional layer, and another batch norm. These general layers are followed by layers specific to the type of autoencoder.

The vanilla autoencoder flattens the convolutional output and projects it to a latent space of size $D$ with a linear layer. We use a value of $k = 8$ and sweep over values of $d = \{16, 64, 256, 1024\}$ for each environment. We use $d = 64$ for the *empty* environment, $d = 256$ for *crossing*, and $d = 1024$ for *door key*, though we note that we do not observe a statistically significant difference in performance for values of $d \geq 64$. The end-to-end baseline uses the same architecture and tuning procedure, but the final hyperparameter values are $d = 64$ for *crossing*, and $d = 1024$ for *door key*.

The FTA autoencoder has the same structure as the vanilla autoencoder, but with an FTA after the final bottleneck layer. The tiling bounds are fixed at $[-2, 2]$ for all cases, except for learning a world model in the *door key* environment, where it is $[-4, 4]$. We sweep over values of $d = \{64, 256, 1024\}$ and the number of tiles, $k = \{8, 16, 32\}$. The sparsity parameters, $\eta$, is set to be the same as the size of the tiles, as is recommended in the original work (Pan et al., 2021). We use values of $d = 64$ and $k = 16$ in both environments.

The VQ-VAE directly quantizes the output of the general layers, so the only other parameters added are the embedding vectors. The number of vectors that make up a latent state is given by $k^2$, and we let $l$ be the number of embedding vectors, resulting in discrete representations of shape $(k^2, l)$. We sweep over values of $k = \{3, 6, 9\}$ and $l = \{16, 64, 256, 1024\}$ for each environment. We use $k = 6$ and $l = 1024$ (for a total size of 6,144) for all environments except for *crossing*, which uses a value of $k = 9$ (for a total size of 9,216).

When designing the experiments, we considered how to construct a fair comparison between the continuous and discrete methods despite the fact that each have different ideal sizes of the latent state, which makes one model bigger than the other. This is a particularly difficult question because it is unclear if we should focus on the size of a representation in bits, or the size of the representation in the number of values used to represent it in a deep learning system. A discrete representation is orders of magnitude smaller than a continuous representation if represented in bits ($9 \times \log_2 1024 = 90$ bits in the *crossing* environment), but takes an order of magnitude more values to represent as one-hot vectors being passed to a neural network ($9 \times 1024 = 9216$ values in the *crossing* environment). Ultimately, we found that answering this question was unnecessary, as the performance of both

methods was limited no matter how large we made the size of the representations. In the *crossing* environment, for example, the performance of the continuous model would not increase even if we increased the size of the latent state from 256 to 9,216 values to match that of the discrete latent state.

## A.4 REINFORCEMENT LEARNING HYPERPARAMETERS

Before running the model-free RL experiments, we performed a grid search over the most sensitive PPO hyperparameters for the continuous model. We swept over clipping values, $\epsilon \in \{0.1, 0.2, 0.3\}$, and the number of training epochs per batch, $n \in \{10, 20, 30, 40\}$. We use the same final PPO hyperparameters for training the RL models with FTA and VQ-VAE latents, which are provided in table 2.

After the sweep over PPO hyperparameters, we also repeated a sweep over the latent dimensions of all of the autoencoders (with the exception of the VQ-VAE, which we found to be robust to a large range of hyperparamers) as described in Section A.3. The vanilla autoencoder and end-to-end baseline use a $d = 256$ dimensional latent space. The FTA autoencoder also uses $d = 256$ dimensional pre-activation latent space with $k = 8$ tiles, forming a 2048-dimensional post-activation latent space. The VQ-VAE uses $k^2 = 36$ latent vectors and $l = 256$ embedding vectors, forming a 9216-dimensional latent space.

Table 2: RL training hyperparameters

| Hyperparameter | Value |
|---|---|
| Horizon (T) | 256 |
| Adam step size | 256 |
| (PPO) Num. epochs | 10 |
| (PPO) Minibatch size | 64 |
| Clipping value ($\epsilon$) | 0.2 |
| Discount ($\gamma$) | 0.99 |
| (Autoencoder) Num. epochs | 8 |

## A.5 EXPERIMENT DETAILS

Table 3: Minigrid environment specifications

| Environment Name | Image Dimensions | Actions | Stochastic | # of Unique States |
|---|---|---|---|---|
| *Empty* | $48 \times 48 \times 3$ | left, right, forward | no | 64 |
| *Crossing* | $54 \times 54 \times 3$ | left, right, forward | yes | 172 |
| *Door Key* | $64 \times 64 \times 3$ | left, right, forward, pickup, use | yes | 292 |

## A.6 MEASURING SPARSITY

In Section 3.3.3, our comparison between multi-one-hot and quantized VQ-VAE representations (Figure 4) resulted in a decisive victory for multi-one-hot representations, which are both sparse and binary. Then in the continual RL setting in Section 4.2, we again see the two sparse representations perform the best. These results suggest that there is an advantage to using sparse representations, but *can we measure the effects of different levels of sparsity?*

In this section, we design an experiment that measures the effects of varying levels of sparsity in the continual RL setting. The most straightforward way to design such an experiment with a VQ-VAE is to change the size of the codebook, which directly controls the level of sparsity. Changing only the codebook, however, also changes the number of the parameters in the model. If we want to measure the effects of *only* sparsity, then we need to control for the size of the model.

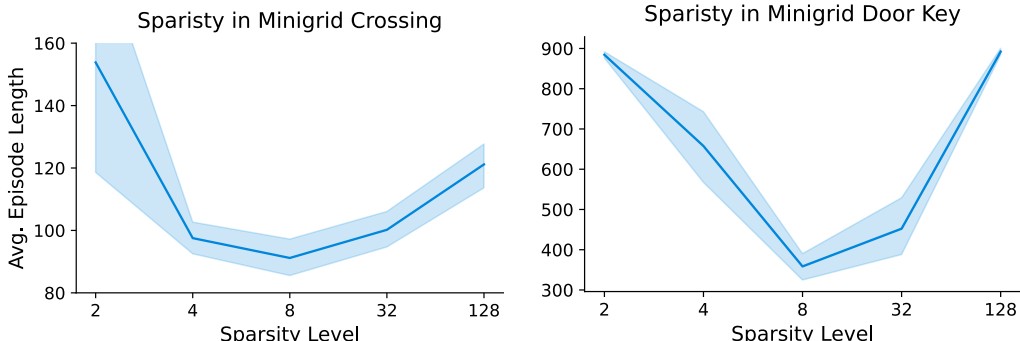

Figure 9: Episode length of a continual RL agent averaged over 15 runs per data point. Lower is better, indicating faster navigation to the goal. All agents use VQ-VAE representations, and the sparsity level indicates the ratio of 0s to 1s in the representation (e.g. a sparsity level of 8 indicates that there are 7 zeros for each one). The shaded region depicts a 95% confidence interval.

In this experiment, we vary the dimensionality of the embeddings, the number of latents, and the size of the codebook all in tandem so that the size of the model stays constant as the level of sparsity changes. At each level of sparsity, we rerun the continual RL experiments as described in Section 4.2 and plot a summary of the results in Figure 9. In the results, we see that **sparsity does help and that there is an ideal amount of sparsity**. In both the *crossing* and *door key* environments, a sparsity level of 8 leads to optimal performance.[3] These results mirror findings from the work on FTA by Pan et al. (2021), which also show sparsity helping up to a certain threshold.

---

[3]Note that the optimal sparsity levels in this experiment do not align with experiments in previous sections because we use a modified architecture that allows us to change the sparsity level more freely.

## A.7 SUPPLEMENTAL WORLD-MODEL MATERIALS

This section contains additional materials that help describe the model training process and results. Algorithm 1 provides pseudo-code for the training algorithm, Figures 10 & 11 visualize the training process, and Figures 12 & 13 visualize distributions of rollouts predicted by the learned world models.

---

**Algorithm 1** Training Autoencoder and World Model

---

$\mathcal{D} \leftarrow$ dataset of transition tuples $(s, a, s')$
Initialize the encoder, $f_{\boldsymbol{\theta}}$, decoder, $g_{\boldsymbol{\phi}}$, and world model, $w_{\boldsymbol{\psi}}$
Set the number of autoencoder training steps, $N$, the number of of world model training steps, $L$, and the number of hallucinated replay steps, $K$

{Training the Autoencoder}
**for** $N$ steps **do**
    Sample transition $(s_0, a_0, s_1) \in \mathcal{D}$
    $\mathbf{z} \leftarrow f_{\boldsymbol{\theta}}(s_0)$
    $\hat{s}_0 \leftarrow g_{\boldsymbol{\phi}}(\mathbf{z}_0)$
    loss $\leftarrow$ MSE$(s_0, \hat{s}_0)$
    Update parameters $\boldsymbol{\theta}$ and $\boldsymbol{\phi}$ with Adam
**end for**
Freeze autoencoder model weights, $\boldsymbol{\theta}$ and $\boldsymbol{\phi}$

{Training the World Model}
**for** $L$ steps **do**
    Sample a sequence of transitions $(s_0, a_0, s_1, a_1, ..., s_K) \in \mathcal{D}$
    $\hat{\mathbf{z}} \leftarrow f_{\boldsymbol{\theta}}(s_0)$
    **for** $k$ in $\{0, 1, ..., K-1\}$ **do**
        $\hat{\mathbf{z}} \leftarrow w_{\boldsymbol{\psi}}(\hat{\mathbf{z}}, a_k)$
        $\mathbf{z}_{k+1} \leftarrow f_{\boldsymbol{\theta}}(s_{k+1})$
        Compute loss between $\hat{\mathbf{z}}$ and $\mathbf{z}_{k+1}$ {cross-entropy for discrete, MSE for continuous}
        Update parameters $\boldsymbol{\psi}$ with Adam
    **end for**
**end for**

---

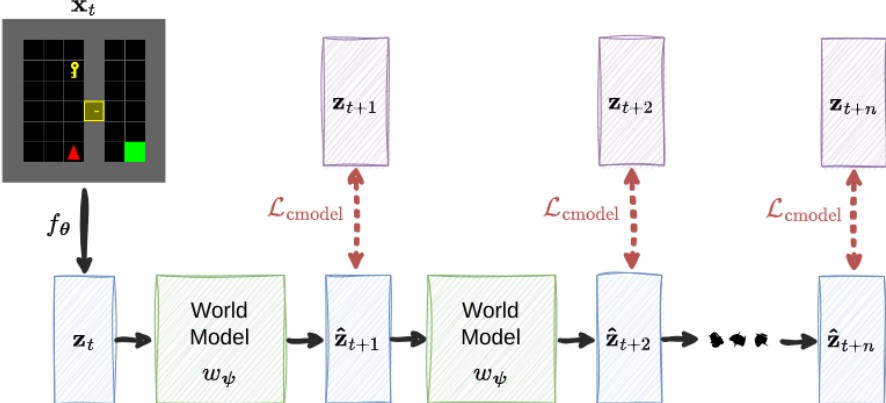

Figure 10: Depiction of a continuous world model training with $n$ steps of hallucinated replay. After encoding the initial observation, the world model rolls out a trajectory of predicted latent states, $\hat{\mathbf{z}}_{t+1}, \hat{\mathbf{z}}_{t+2}, \ldots, \hat{\mathbf{z}}_{t+n}$. Actions from a real trajectory are used during training, but are excluded in the depiction to avoid clutter. The loss at each time step is calculated as the mean square error between the hallucinated latent state $\hat{\mathbf{z}}_{t+i}$ and the ground-truth, $\mathbf{z}_{t+i}$. This method is called hallucinated replay because the entire trajectory after the first latent state is hallucinated by the world model.

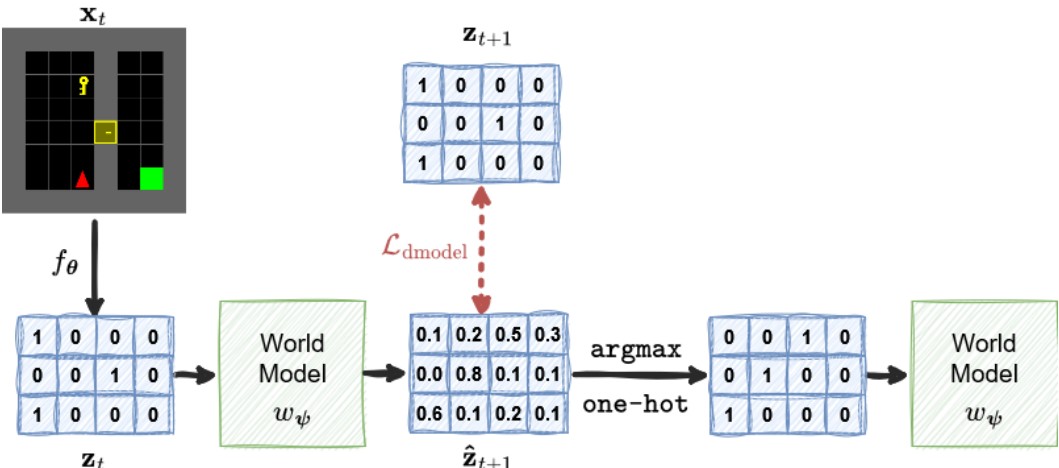

Figure 11: Depiction of a single step of discrete world model training and the subsequent discretization of the latent state. The observation $\mathbf{x}_t$ is encoded to produce latent state $\mathbf{z}_t$, which is passed to the world model to sample the logits $\hat{\mathbf{z}}_{t+1}$ for a following state. The predicted next state logits $\hat{\mathbf{z}}_{t+1}$ are compared to the ground truth state $\mathbf{z}_{t+1}$, which is constructed from the corresponding ground-truth observation: $\mathbf{z}_{t+1} = f_{\boldsymbol{\theta}}(\mathbf{x}_{t+1})$. Before the world model can be reapplied, the latent state logits must be discretized with an `argmax` operator and converted to the one-hot format.

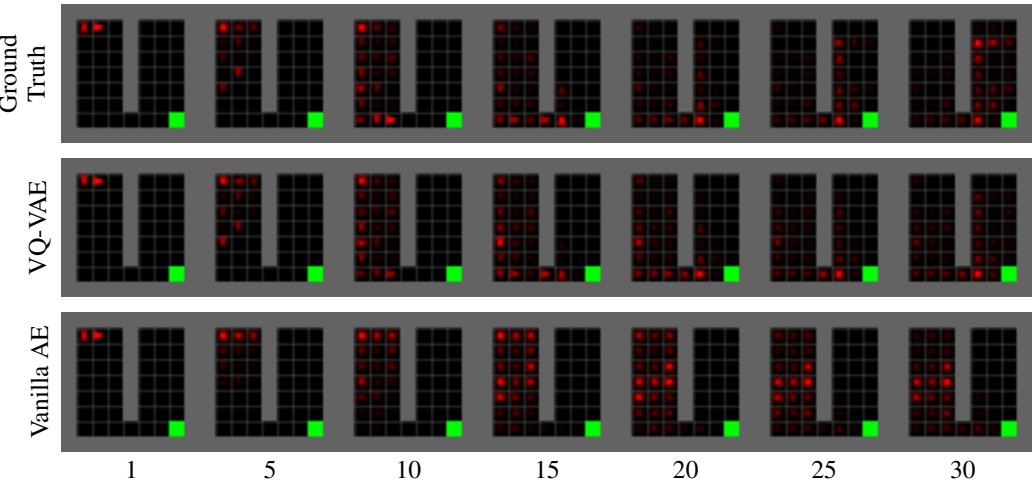

Figure 12: Comparison of rollouts predicted by different world models in the *crossing* environment. Each row visualizes the state distributions throughout rollouts predicted by different world models, with the x-axis giving the step in the rollout. The ground-truth row depicts the state distribution over rollouts as a policy that explores the right side of the environment is enacted in the true environment. Predicted observations are averaged over 10,000 rollouts. Being closer to the ground-truth indicates a higher accuracy.

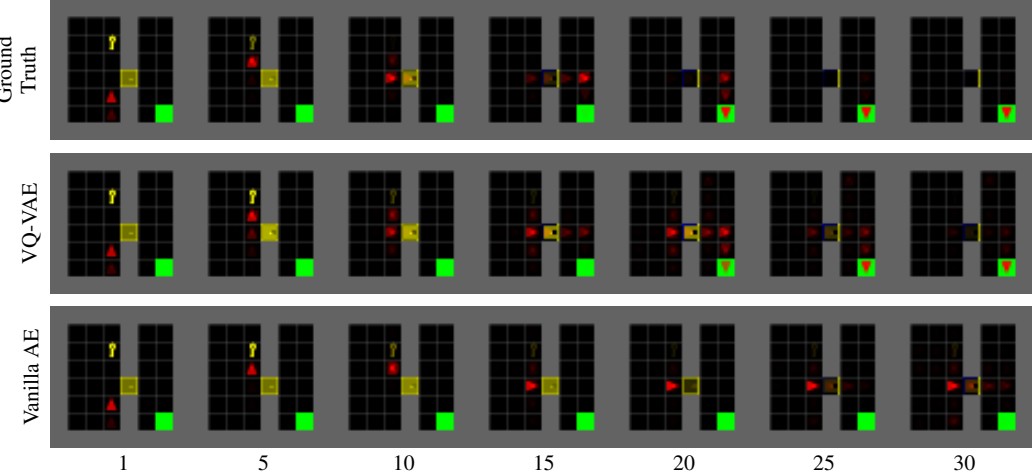

Figure 13: Comparison of rollouts predicted by different world models in the *door key* environment. Each row visualizes the state distributions throughout rollouts predicted by different world models, with the x-axis giving the step in the rollout. The ground-truth row depicts the state distribution over rollouts as a policy that navigates to the goal state is enacted in the true environment. Predicted observations are averaged over 10,000 rollouts. Being closer to the ground-truth indicates a higher accuracy.

A.8 SUPPLEMENTAL RL MATERIALS

This section contains additional materials that help describe the RL training process and results. Algorithm 2 provides pseudo-code for episodic and continual RL training. Figure 14 shows different environment variations used in the continual learning setting. Table 4 provides analytical data on continual RL performance. Figure 15 plots the reconstruction loss of the autoencoder during episodic RL training. And lastly, Figure 16 depicts the full results of the continual RL runs starting from the first timestep.

---

**Algorithm 2** Reinforcement Learning Training Process

---

Initialize the encoder, $f_\theta$, and decoder, $g_\phi$
Initialize the policy and value networks, $\pi_\psi$ and $V_\psi$, with combined parameters $\psi$
$\mathcal{D} \leftarrow \emptyset$ {Dataset of observations}
Set number of interaction steps, $N$, batch size, $B_0$, autoencoder epochs, $L$, PPO epochs K, PPO start step $P$, and autoencoder batch size, $B_1$
For continual learning experiments, specify environment change frequency, $C$

**while** number of interactions is less than $N$ **do**
    Enact policy $\pi_\psi$ in the environment to obtain a batch of $B_0$ transition tuples
    **if** interaction step $\geq P$ **then**
        Using the online data, perform $K$ epochs of PPO updates on parameters $\psi$
    **end if**
    **for** L steps **do**
        Sample a batch of observations $(s_0, s_1, ..., s_{B_1}) \in \mathcal{D}$
        Apply the autoencoder and calculate the reconstruction loss
        Update parameters $\theta$ and $\phi$ using Adam
    **end for**
    **if** doing continual learning and $C$ interaction steps have passed **then**
        Randomize the environment
    **end if**
**end while**

---

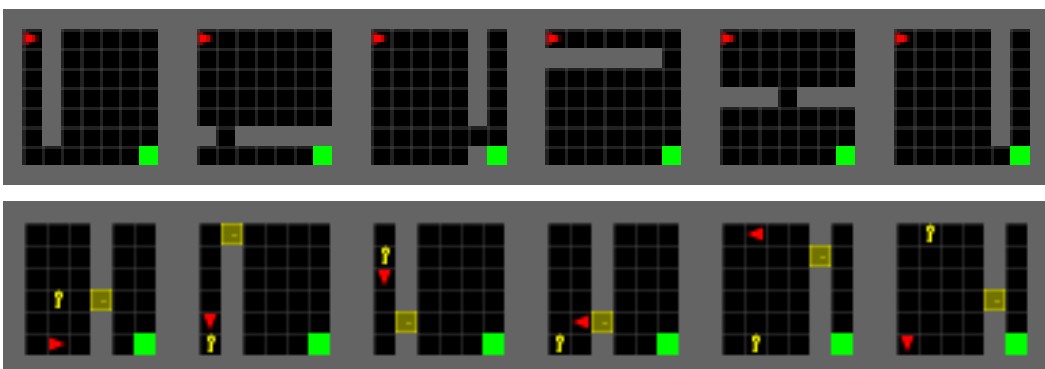

Figure 14: The top row depicts random initializations of the *crossing* environment, and the bottom that of the *door key* environment. Each time the environment changes, the positions of all internal walls and objects are randomized, with the exception of the agent position in the *crossing* environment and the goal in both environments.

Table 4: RL performance per environment layout (95% CI)

| Latent Type | *Crossing* Reward | *Door Key* Reward |
|---|---|---|
| End-to-End | $28 \pm 5$ | $14 \pm 2$ |
| Vanilla AE | $382 \pm 33$ | $866 \pm 94$ |
| FTA AE | $574 \pm 57$ | $1033 \pm 130$ |
| VQ-VAE | $\mathbf{674 \pm 21}$ | $\mathbf{1324 \pm 64}$ |

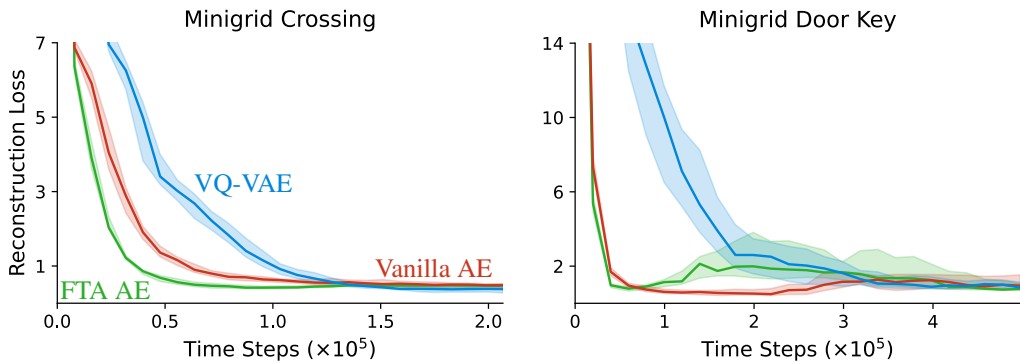

Figure 15: Median reconstruction loss of the autoencoder during episodic RL training. The autoencoder is trained on observations randomly sampled from a buffer that grows as the RL training progresses. Lower is better, indicating a better reconstruction of the input observation. The plot depicts a 95% confidence interval around the median over 30 runs. We plot the median of this metric as there are a few outliers that drastically skew the average. The VQ-VAE in particular exhibits the highest variance in reconstruction loss, but this does not seem to hinder the representation's performance in the RL setting.

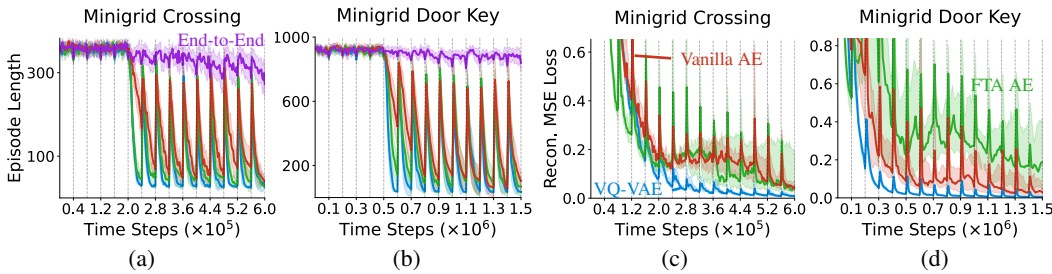

Figure 16: (a-b) Mean agent performance as the environments change at intervals indicated by the dotted, black lines. Lower is better. (c-d) Median encoder reconstruction loss. Lower peaks mean the representation generalizes better, and a quicker decrease means the autoencoder is learning faster. Overall, a lower reconstruction loss is better. (a-d) Results are averaged over 30 runs and depict 95% confidence intervals. Performance is plotted after an initial delay to learn representations, after which all methods are trained with PPO.

