# OpenReview forum: "Harnessing Discrete Representations for Continual Reinforcement Learning"
_ICLR.cc/2024/Conference — Submitted to ICLR 2024_

### Official Review · Reviewer_KUuk · 2023-10-30

**Soundness:** 3 good
**Presentation:** 3 good
**Contribution:** 2 fair
**Rating:** 5
**Confidence:** 4

**Summary:**

This paper conducts an empirical analysis to gain insight into the success of discrete world models in reinforcement learning. It conducts a series of experiments to disentangle the effects of sparse vs dense and continuous vs discrete representations on a set of MiniGrid tasks. It concludes that sparsity, rather than simple discretization, plays a significant role in the improved learning dynamics enjoyed by the VQ-VAE representation in both single-task and continual reinforcement learning tasks.

**Strengths:**

- This is a well-written paper which clearly states its problem of interest, methodology, and conclusions.
- The paper studies an interesting and important topic that has previously been under-explored
- The experimental setting is easy to follow and interpret. The architectural assumption of using a fixed set of encoders and switching between task-specific "heads" is a nice way to control for different architectural variants, giving an apples-to-apples comparison between different design choices.
- The snap-to-grid and continuous tile coding strategies provide an insightful continuum between discrete and continuous, and between sparse and dense representation types.
- Figure 5 provides a particularly insightful investigation into the effect of sparsity.

**Weaknesses:**

- The paper only considers gridworld tasks, and it is hard to say whether the findings will generalize to more visually rich and complex planning tasks. A paper which only tells the reader something about representation learning in gridworlds presents a much less useful and interesting contribution than one whose insights are expected to generalize to a broader class of environments that are of interest to practitioners. This is the main issue which is lowering my score.
 - The RL task is extremely simple for the fixed representation: while the paper does point out that the discrete representations converge 2-4x faster than the representations obtained from the continuous reward model, both representations seem to converge almost immediately
  - The error bars on many of the figures are quite large, and there were some strange artifacts, in particular in Figure 3 where going from 512 to 1024 dimensions dramatically increased the error of one of the models, which are not explained. Given the relatively narrow scope and simplicity of the domains under consideration, this increases my concern about how the findings in this paper will generalize to other environments. I understand that evaluating on many tasks requires significant computational resources, but I worry that the results in this paper might be telling us more about the MiniGrid environment than about the general properties of the representations studied.
  - Relatedly, the gap between FTA and VQ-VAE is much larger in one environment than the other, suggesting that there may be domain-specific properties that influence the relative performance of different representations. It would be useful to have a broader range of evaluation environments, even if these are limited to the minigrid setting to see whether this is the case in general.

**Questions:**

- In the description of the continual RL setting, it wasn't clear to me whether the representations were trained on a variety of shuffled layouts or only a single one. It would be very surprising to me if training a transition model on a single layout led to features that generalized so effectively (with apparently no degradation in performance on OOD inputs), but if this was the case it would merit further discussion.
- One important factor influencing the usefulness of model-based representation learning is whether the data collected for the pretraining phase covers the state space. Since the transitions are mostly collected from random policies, this seems like it could be a significant issue in a domain like atari or mujoco, where a random policy samples a very limited state distribution. How dependent are the results in this paper on the pretraining distribution used for the learned representations?
- Minor: the paper briefly mentions that it will focus on sample-based models, but the reasoning behind this choice isn't fully explained. Is there a reason why expectation models would not be suitable for representation learning? Or is this motivated by the desire to replicate the setting used in prior empirical work?

---

> ### Author Response · Authors · 2023-11-18
>
> Thanks for reviewing our work!
> &nbsp;
>
> &nbsp;
>
> **Weaknesses**
>
> **[W1]** The use of only Minigrid environments raises concerns about the generality of the results.
>
> **[A1]** We posted an in-depth, public comment addressing this point.
> &nbsp;
>
> &nbsp;
>
> **[W2]** The RL tasks is too simple for the fixed representation.
>
> **[A2]** As you point out, the RL task is not too difficult if a good representation is already learned. Our main goal in the episodic RL section is to build up motivation for doing continual learning where these differences can be significant. Looking at the Door Key environment, the VQ-VAE-based agent takes ~80k steps to converge, and the AE-based agent takes ~250k steps. For an agent that you only ever train once this may not matter, but efficiency is extremely important for an agent that needs to adapt to a constantly changing environment—i.e. continual RL. Our long term vision is to have continually learning agents that are much more efficient than both of these (imagine an agent that could learn in just several episodes!). But for now, we think that saving 170k steps is a good step in the right direction.
> &nbsp;
>
> &nbsp;
>
> **[W3]** Some of the error bars are large, with a particularly noticeable artifact in Figure 3.
>
> **[A3]** The larger error bars for the end-to-end method in Figure 3 are produced by a bimodal distribution caused by a few diverging runs. We believe that some of the end-to-end runs diverge because the model has to backpropagate multiple loss terms through both the transition model and encoder over multiple time steps. In contrast, the other methods train the autoencoder and transition models separately and do not backpropagate over multiple time steps, which makes the optimization process much simpler. Your comment has made it apparent that our presentation of the data here was not clear enough. To fix this, we are switching the figure to plotting the median and adding a further explanation to the caption. We will upload a new manuscript within the next 24 hours that reflects these changes.
>
> Our work aims to be a thorough study that improves the understanding of this area, and we think that reporting all results, including "not clean” ones, is an important part of helping others understand the realistic state of these problems. In regards to error bars, we are very careful to only make claims that are supported by statistically significant results throughout the paper, and the largest error bars primarily seem to be from standard RL (end-to-end) rather than the methods we investigate.
> &nbsp;
>
> &nbsp;
>
> **[W4]** The gap between FTA and VQ-VAE is much larger in one environment than the other, which may suggest domain-specific properties are at play.
>
> **[A4]** Assuming you are talking about the model learning setting (if not, let us know and we can follow up), the difference in scale is the result of our evaluation metric. The objective in the crossing environment causes the agent to spread out, whereas the objective in the door key environment concentrates the agent into just a few states. Because of this, the KL divergence is much higher in the door key environment when the agent is off by a few steps. This can be seen in Figures 11 and 12, where there is more overlap in the crossing environment.
> &nbsp;
>
> &nbsp;
>
> **Questions**
>
> **[Q1]** Are the representations trained on multiple layouts?
>
> **[A1]** The representations are trained on multiple layouts. To clarify how the continual learning training works, both an autoencoder and a policy are continually trained in tandem as new data comes in, with the exception of an initial delay before the policy starts to learn. We will have an updated manuscript uploaded within the next 24 hours that clarifies this.
> &nbsp;
>
> &nbsp;
>
> **[Q2]** How dependent are the results on the distribution of the pretraining data?
>
> **[A2]** In the continual learning section, the representations are trained continually as the policy changes even after the initial pretraining period, so the representation changes as the distribution shifts. We found this to work well. We also tried experiments without the pretraining phase. This also worked, but we found the RL performance to be less stable, suggesting that there is a plasticity issue with the policy not being able to update to the changing representations. We think this could be solved by a number of recent interesting works on the stability-plasticity dilemma (e.g. [Dohare et. al 2023](https://arxiv.org/abs/2306.13812), [Nikishin et. al, 2023](https://openreview.net/forum?id=O9cJADBZT1)), but we have yet to test it.
> &nbsp;
>
> &nbsp;
>
> **[Q3]** Why use sample-based models over expectation models?
>
> **[A3]** We use sample-based models primarily to build on top of prior work (e.g. DreamerV3), but also because expectation models tend to be more limited, as they lose information about stochastic environments.

---

> > ### Comment · Reviewer_KUuk · 2023-11-19
> >
> > Thanks to the authors for their response, which have addressed several of my concerns. I've provided my follow-up concerns and questions below.
> >
> > W1.  I agree that the choice domain allows for faster experimentation and greater statistical significance of the paper’s  empirical results, but the limitation to only two tasks even within the mini grid domain is extreme. When performing an analysis of an empirically-observed phenomenon in a simplified setting, it is important to be able to show that the findings of the analysis in the simplified setting also hold in the originally-observed setting. The biggest barrier to increasing my score is the limited evidence that findings in two mini grid tasks tell us something interesting about general properties of discrete world models. Showing how these results change if you increase the size of the grid world, vary the stochasticity of the transitions, or increase the visual complexity/prevalence of distractors in the environment, would provide some indication that the findings presented in this paper are not dependent on some shared property of the two tasks.
> >
> > W3. I appreciate the clarification regarding artifacts, but if a significant number of runs are diverging then this is an important factor in algorithm’s performance and raises additional questions. Is the divergence on high-dimensional embeddings a product of poor hyperparameter transfer, or is it due to an inherent instability in the approach?
> >
> > Q1: I realized that part of my confusion about this figure came from the X-axis offset in Figure 6, which does not show performance from the start of training but rather only from the point where learning curves would started to go down in the single-task setting. Can the authors show the full trajectory so that we can see what the early training period looks like? Otherwise it looks like the world model methods where somehow able to converge faster on the continual task than the single task version of the environments.
> >
> > Q2: I am specifically concerned about settings where the nonstationarity induced by policy improvement results in significant changes to the dynamics and structure of the environment (for example, reaching a new level in a video game). The mini grid environment is nice in that the small number of states means that a random policy will attain good coverage. Even if the layout is re-randomized, the overall structure in the environment is preserved, and so I’d expect that even a random exploratory policy in a different layout will reveal a lot of useful information for future layouts. I don’t think this holds for all environments, and would want to see robustness to a more ‘pathological’ continual learning problem to be convinced that the world models studied are meaningfully robust to nonstationarity.
> >
> > Errata:
> >
> > - I noticed in Figure 5 that there is a discrepancy between the x-axis scales in subplots (a) and (c). Which one is correct?

---

> > > ### Author Response · Authors · 2023-11-23
> > >
> > > Thank you for the follow-up response. Addressing your points:
> > >
> > >
> > > **[W1]** Unfortunately, we were unable to prepare additional experiments in time. While experiments in different environments would certainly be nice to have, there is existing prior work that experiments with discrete representations (including comparisons to continuous representations) in Atari ([Hafner et al.,  2020](https://arxiv.org/abs/2010.02193)), and our results align with the findings in that work. Standalone results in only two environments would generally be a low bar for us too, but the alignment of our results with previous work is what gives us more confidence in the results.
> > >
> > > **[W3]** The divergence on high-dimensional embeddings is likely a product of an inherent instability in the approach, as we separately fine-tune each of the models with different latent dimensions. Backpropagating through multiple timesteps simply makes optimization much harder, and having multiple targets further complicates that process. While having diverging runs is not ideal, the conclusions of the experiment would not change whether or not the runs existed, as the problem only applies to the standard RL baseline.
> > >
> > > **[Q1]** We are happy to provide the full data. Figure 16 in the appendix depicts the data starting from timestep 0. We do not show the early part of training in the main body of the paper because no RL training is happening, and hence, the performance is not changing. Only the autoencoder is trained at the start of the process, and RL updates only begin after a delay. An overview of the training algorithm is described in Algorithm 2 in the Appendix.
> > >
> > > **[Q2]** This is a very interesting direction to explore, as it really grasps at the heart of continual learning. In the future, we would love to see if our method can help adapt more quickly in those more significant changes. Even in the case of Minigrid, the changes can be quite significant when you consider that the model has only ever seen between 1 and 15 variations of the environment in its lifetime. This is reflected in spikes in reconstruction error seen in subplots (c) and (d) of Figure 6. That being said, there are environments that will undergo even more drastic changes as you mentioned. Perhaps testing on a domain like Atari, going from one Atari game to another, would be an interesting direction for future work.
> > >
> > > **[Errata]** Thank you for catching this. Subplot (a) was correct, and we have fixed subplot (c) accordingly.

---

### Official Review · Reviewer_CBeE · 2023-10-30

**Soundness:** 2 fair
**Presentation:** 3 good
**Contribution:** 3 good
**Rating:** 5
**Confidence:** 4

**Summary:**

The authors observe that one of the must fundamental factors in RL is how agents represent observations, and in particular, the form of these representations. While discrete representations have found success recently, there has not been a systematic study of dense vs. sparse vs. discrete representations - which is the focus of this work. The authors implement agents with these 3 kinds of representations in model-based, model-free, and continual RL settings, and perform several experiments to disentangle the effects of these representations, focusing on the better-performing discrete representations, and hence present a few novel findings.

**Strengths:**

I believe this work is very well motivated - to the best of my knowledge there has not been a third-party examination of the role of these different types of representations within RL, and it is important for the community to understand such a fundamental issue better. And beyond simply trying dense, continuous vectors, and discrete vectors, the authors also try sparse, continuous vectors, which are an understudied but potentially valuable form of representations. They present a range of interesting studies, and do address an important tradeoff - discrete representations can have their advantages, particularly in modelling rarer parts of the distribution/not focusing capacity on just the most common events, but can also require more learning.

**Weaknesses:**

I believe a crucial oversight in this work is that, while the authors focus on the effects of types of representations, they failed to take into account the representational capacity of the representations, or rather, the regularisation. There is a considerable amount of literature on "disentanglement" in NNs - particularly VAEs - showing that regularisation is key to models learning to separate different latent factors in data generation process. So although the hidden size of the bottleneck in AEs does reduce representational capacity, I believe that without further regularisation on the continuous latent space - for example, using a β-VAE - this is not taken into account properly.

There is also another major caveat with these findings, which is that all experiments were conducted on the Minigrid environment, which is a gridworld, and hence one might expect discrete representations to better match the fundamentally discrete objects in the environment. Although this doesn't invalidate the findings of this work, I believe the authors need to address this shortcoming. Ideally, the authors would show that their findings hold in more complex environments - DeepMind Lab is typically used in this regard, as it also has discrete objects and a grid-like environment, but it is 3-dimensional and observations are provided in first-person RGB views.

**Questions:**

I do not have any particular questions for the authors. I appreciate the thorough and clear implementation details in the appendix.

**Edit:** I have read the other reviews, the authors' feedback, and the updated paper. Whilst I commend the authors for their clarifications, I believe that the weaknesses raised are still not sufficiently addressed, and will not therefore not update my rating. However, I am enthusiastic about this work, and believe that with improvements it would be a valuable contribution to the RL community.

---

> ### Author Response · Authors · 2023-11-18
>
> Thank you for reviewing our work! Addressing your comments:
> &nbsp;
>
> &nbsp;
>
> **Weaknesses**
>
> **[W1]** Concerns about representational capacity and the lack of regularization-focused baselines, particularly VAEs, in the study.
>
> **[A1]** Your first comment brings up two great points, one on accounting for representational capacity when comparing methods, and the other on regularization-focused baselines, namely VAEs. The comment on VAEs is actually quite prescient, as we had a very similar idea and started with a VAE baseline earlier in the project. After a number of experiments, however, no set of VAE hyperparameters was able to approach the performance of a simple vanilla AE for model learning. When inspecting the runs, we found that the regularization imposed by the VAEs made it too difficult for the model to reconstruct observations. Given the huge gap in performance, we opted to scrap the VAE and perform the remainder of the experiments with vanilla AEs.
>
> It is possible that a VAE may work better in the continual RL setting, but our previous results in the model learning setting lowered our expectations. We are currently working on preparing an experiment in that setting, but we do not know if we will have the results in time, so we wanted to post an initial response to allow time for any additional comments. We will follow up here when we have updates on this front.
>
> We focused on the part of your comment about VAEs as that seemed to be your primary concern, but we would also be happy to discuss comparisons between representational capacity in further depth as a follow up if desired.
> &nbsp;
>
> &nbsp;
>
> **[W2]** Usage of only MiniGrid environments in experiments.
>
> **[A2]** On the topic of your second point, we posted an in-depth, public comment in regards to the use of only MiniGrid environments.
> &nbsp;
>
> &nbsp;

---

> > ### Comment · Reviewer_CBeE · 2023-11-18
> > **VAEs for Model Learning**
> >
> > To preface, I empathise that tuning models and running experiments is a laborious process. At the very least, the paper would be improved by explicitly acknowledging this concern and explaining that there were difficulties in tuning a VAE for model learning.
> >
> > > When inspecting the runs, we found that the regularization imposed by the VAEs made it too difficult for the model to reconstruct observations.
> >
> > I believe the simplest solution would be to put a weight β < 1 on the KL loss component, finding a value that allows sufficient reconstruction/performance with some regularisation on the latent space.
> >
> > However, simply using a Gaussian prior would invite another confounder - whilst an AE has no restrictions on representations, a vanilla VAE is softly constrained towards a unimodal latent space (unlike the other methods examined in your paper). In which case using a GMM prior would make for a fairer comparison. Such a model would, like the other methods, explicitly encourage disentanglement into multimodal latent factors, whilst retaining completely continuous representations.

---

> > > ### Author Response · Authors · 2023-11-22
> > >
> > > Thank you for the response! We have spent a good deal thinking on this, and these are our main thoughts:
> > >
> > > 1. We agree that a comparison to a VAE with a GMM prior would provide valuable insights, though it is unfortunately not something we have time to properly tune and run. As you suggested, we have added a note to the background on baselines that acknowledges our attempts at training a VAE baseline. It also includes a mention that such a method may be improved by variations like beta-VAEs or VAEs with GMM priors. Thank you for the suggestion. We do think the addition makes the paper better, and a look into whether these different types of VAEs could help in continual RL would be an interesting and novel direction for future work! These changes will be uploaded tomorrow as part of our final draft.
> > >
> > > 2. The discussion on the lack of a regularized continuous representation baseline seems to be very focused on the benefits of disentanglement, but our results in section 3.3.3 and with FTA suggest that there is actually something else going on. In section 3.3.3 we compare two representations that are both produced by the same VQ-VAE, and hence should have the same level of disentanglement, yet the sparse, binary one performs much better than the continuous one. FTA also has no explicit design choice that encourages disentanglement, yet we see it perform nearly as well as the VQ-VAE throughout the RL experiments. Disentanglement in the representation space may help in the tasks we test, but we believe our results actually point to a different, interesting phenomenon.
> > >
> > > 3. Even without a VAE comparison, we believe that our work still delivers on our three stated contributions:
> > > >   - Elucidating multiple ways in which discrete representations have likely played a key role in successful works in model-based RL.
> > > >   - Demonstrating that the successes of discrete representations are likely attributable to the choice of one-hot encoding rather than the “discreetness” of the representations themselves.
> > > >   - Identifying and demonstrating that discrete and sparse representations can help continual RL agents adapt faster.
> > >
> > > Whether these contributions warrant publication is, of course, a decision in the hands of the reviewers, but they are interesting findings that stand even without additional continuous representation baselines.

---

### Official Review · Reviewer_Ys23 · 2023-11-03

**Soundness:** 3 good
**Presentation:** 3 good
**Contribution:** 3 good
**Rating:** 8
**Confidence:** 3

**Summary:**

This paper takes a deeper look at use of discrete representations rather than continuous ones, performing empirical studies to develop a potentially deeper understanding of when these representations are useful. The authors explore how helpful these representations are both when learning environment models and when learning model-free policies. The authors find that discrete representations are more helpful when capacity is limited and propose it as a potential solution for continual RL scenarios for this reason. As predicted, their continual learning experiments also demonstrate performance gains when using discrete representations.

**Strengths:**

- I enjoyed reading this paper and found it mostly very clear and easy to follow.
- I think that the paper asks an important question. I also often wonder about the benefits achieved when using discrete representations.
- The conclusion of the experiments feels very believable to me, even if the empirical results aren't comprehensive enough to fully support the generality of the claims.
- I find the argument spelled out for use of discrete representations in continual RL to be an important contribution to the discourse around architectures in this research area, making the paper potentially interesting to a pretty broad audience.

**Weaknesses:**

- The experiments do look at scale, but the architectures / domains considered are very basic, so it is unclear how general these results are and how applicable they are to real-world scenarios.
- Also the breadth of experiments is not very large in terms of domains when we consider that providing a comprehensive empirical study is a stated goal of the paper.
- The authors could potentially make the paper better by providing a bit more in terms of direct empirical proof of their interpretations i.e. the role of sparsity,

**Questions:**

1. The authors mention a few times that there is a connection between the success of discrete representations in continual RL and representation sparsity. Did you try actually measuring this sparsity? I guess the size of each one-hot dictates this sparsity level, so it could be interesting to understand the connection between this and continual learning performance.

2. In section 3.3.3 I am a bit unclear about what the conclusion is for these experiments. What are some of the insights you believe they present? Is this actionable in a general sense?

---

> ### Author Response · Authors · 2023-11-18
>
> Thanks for reviewing our work! Addressing your comments:
> &nbsp;
>
> &nbsp;
>
> **Weaknesses**
>
> **[W1]** It is unclear how applicable these results are to real world problems.
>
> **[A1]** We posted a public comment in regards to the use of only Minigrid environments with an in-depth discussion on what that means for generalization to more complex problems.
> &nbsp;
>
> &nbsp;
>
> **[W2]** The breadth of the experiments in not very large.
>
> **[A2]** The other comment mentioned above is also related, but in general, we erred on the side of thoroughness for this work. Keeping to the same domain allowed us to look at many properties of these representations while limiting the number of moving pieces. That also allowed us to perform many runs and have high confidence in the validity of our results.
> &nbsp;
>
> &nbsp;
>
> **[W3]** The authors could potentially improve the paper by providing more empirical results on sparsity.
>
> **[A3]** We are combining the response to your final point under weakness with the response to the first question, as they are closely related.
> &nbsp;
>
> &nbsp;
>
> **Questions**
>
> **[Q1]** The authors mentioned connections to sparsity, but did they actually run any experiments that measure this?
>
> **[A1]** After reading your comment on the limited amount of empirical proof related to sparsity, I ctrl+f searched over the paper and realized how little it was mentioned – thank you for catching this oversight! We are making a couple changes on this point:
> 1. We are adding some extra discussion to section 3.3.3, as the results of the experiment from that section directly support our interpretations on sparsity. This interpretation of the results was previously missing.
> 2. As you point out in your comment, the question of how the sparsity affects continual learning performance is very interesting, so we are currently running some additional experiments to find out. The original paper on FTA ([Pan et al., 2021](https://arxiv.org/abs/1911.08068)) actually runs an experiment just like this (Figure 7a), where they vary the level of sparsity. They find that it helps, but there is an optimal amount of sparsity, and too much or too little sparsity doesn’t perform as well. The preliminary results of our currently running experiments are looking to draw very similar conclusions. We are posting an initial response now to allow time for responses, but we will follow up again after we have added the new section on sparsity.
>
> Our interpretations are also influenced by many other works that have found similar connections to sparsity (e.g., [Ratitch & Precup, 2004](https://link.springer.com/chapter/10.1007/978-3-540-30115-8_33), [Liu et al., 2018](https://arxiv.org/abs/1811.06626), [Mocanu et al., 2018](https://www.nature.com/articles/s41467-018-04316-3)).
> &nbsp;
>
> &nbsp;
>
> **[Q2]** What are the takeaways from section 3.3.3?
>
> **[A2]** The original takeaway from section 3.3.3 was that the implicit design of representing discrete latents with a one-hot encoding scheme is an incredibly important (and often overlooked) choice. Our experiment shows that it is not necessarily the “discreteness” of the representation as a whole that matters, but rather the sparse and element-wise binary nature of the representation that matters. We are adjusting the section to make this clearer, and also to discuss how this is evidence for the importance of sparsity.
> &nbsp;
>
> &nbsp;
>
> We are still working on some of the changes we mentioned in this response, but all of the changes (aside from the new experiments) will be completed and uploaded with a new manuscript within the next 24 hours.

---

> ### Author Response · Authors · 2023-11-23
>
> We are posting an update to let you know that we ran some final experiments on sparsity, and the final draft of the paper we uploaded has a new section on sparsity in the Appendix (A.6). To summarize the results, we found a clear, positive correlation with sparsity and continual learning performance up to a certain point. However, going beyond that point and having too much sparsity again becomes harmful.

---

### Official Review · Reviewer_Cvsu · 2023-11-04

**Soundness:** 3 good
**Presentation:** 4 excellent
**Contribution:** 3 good
**Rating:** 6
**Confidence:** 4

**Summary:**

This paper studies the use of discrete representations in reinforcement learning, specifically in three different setup: model-based, model-free and continual RL. The discrete representations are learned using vector quantized variational autoencoder (VQ-VAE), whereas the continuous representations are learned using the canonical autoencoder and an autoencoder which uses Fuzzy Tiling Activation (FTA).   Using various environments based on minigrid, their work showed that models learned with discrete representations approximate the ground truth distribution when compared to models learned with continuous representations using KL-divergence. Additionally, unlike their counterparts, agents using models learned with discrete representations exhibit better learning efficiency overall.

**Strengths:**

The paper is well-written and mostly easy to understand. The research question is a relevant one for the reinforcement learning community. There have been several uses of discrete representations in previous work as presented by the authors but yet to be studied extensively. To the best of my knowledge, the main relevant work is also cited in the paper. The setup and the aim of the experiments are well-defined and comprehensive. Using simpler experiments such as minigrid enables to author to perform in-depth analysis and gave clear insights of the hypothesis. A broad range of scenarios ranging from model-based to model-free and continual RL were considered and the consistency shown through these experiments added assurance to their findings. The figures in the appendix, for example figure 9,10, 11 and 12 are well presented and helped me to understand their work better, which I greatly appreciate.

**Weaknesses:**

My only slightly major concern on this work is that all experiments are done on the minigrid environments. The pixel observations for this environment are simple which might have contributed to the easier process of discretization. I think to improve the quality of the paper, it will be important to see if the same conclusions can be made when using more visually complex environments, such as Atari 100k, which is also performed in the dreamer paper.

Minor concerns:
1. Will be great if the authors can add a pseudocode of the algorithm in the appendix to improve the clarity
2. Figure 14: Although the trend of the result is clear, this should be plotted with mean and standard deviation to improve readability.
3. Some spacing gaps in the appendix can be reduced, for example using vspace in latex.

**Questions:**

1. In section 3.3.2, the authors mentioned that “In the plot, an interesting pattern emerges: the performance of all methods become indistinguishable beyond a certain size of the world model”. Perhaps this might hint that the environment is too simple?
2. In section 3.3.3, the authors mentioned that “It is alternatively possible that the discrete world model performs better simply because the VQ-VAE learns different information that is more conducive to world modeling.” What other different information do the authors think VQ-VAE is learning?

---

> ### Author Response · Authors · 2023-11-18
>
> Thank you for the review! First, addressing the weaknesses and minor concerns you voiced.
> &nbsp;
>
> &nbsp;
>
> **Weaknesses**
>
> **[W1]** Do these results scale to more visually complex environments?
>
> **[A1]** We posted an in-depth, public comment in regards to the use of only Minigrid environments.
> &nbsp;
>
> &nbsp;
>
> **[W2-4]** Minor concerns around pseudo-code, presentation of Figure 14, and spacing gaps in the appendix.
>
> **[A2-4]** We are working to update the manuscript to address all of the minor concerns and will have a new revision posted within the next 24 hours. Thank you for the suggestions.
> &nbsp;
>
> &nbsp;
>
> **Questions**
>
> **[Q1]** Are the methods performing the same in section 3.3.2 because the environment is too simple?
>
> **[A1]** Yes, this is a great catch and precisely what we aim to show with section 3.3.2. It doesn’t matter what which of our representation methods is used when the environment is simple because all of the methods will perform near-optimally. As the environment becomes more complex, however, discrete representations shine because they are able to learn more with less modeling capacity. It is this result that motivates the use of these representations in continual learning, as continual reinforcement learning is all about how to maximize reward when the agent cannot perfectly model the world (because the world is continually changing from the agent’s perspective).
> &nbsp;
>
> &nbsp;
>
> **[Q2]** What different information do the authors think the VQ-VAE may be learning in section 3.3.3?
>
> **[A2]** We really mean to point out the following: latent spaces are defined both by the information they represent (information) and by the way in which they represent that information (representation). Our goal is to measure how representation alone affects performance, so we ideally want to control for the informational content of the latent space. In section 3.3.3, we do not really expect that the autoencoders are learning different information, but we need to check this possibility to be thorough. We have reworded part of the section to make this clearer.

---

### Author Response · Authors · 2023-11-18
**Addressing criticisms on scale**

We are making this general post to (1) discuss why we use Minigrid for all of our experiments, and to (2) address concerns regarding generalization to other domains. As each reviewer commented on this, this post is intended for all of the reviewers.
&nbsp;

&nbsp;

**Why we used Minigrid**

Our decision to use Minigrid as a testbed was based on our goal to perform a more extensive empirical evaluation surrounding the successful use of discrete representations, and for that, we needed to be able to run large amounts of experiments and have a high level of control over our environment. Using Minigrid allowed us to go beyond reporting only performance curves but also the compare the state distributions induced by the different models (Figure 2), to vary the tasks in a way that we could simulate a continual learning problem (Figure 6), and even visualize distributions over rolled out trajectories (Figures 11 & 12).
&nbsp;

&nbsp;

**Addressing the criticisms**

The concerns of the reviewers seem to be roughly split into two categories:
&nbsp;

&nbsp;

*1. It is unclear whether discretization will be as easy, or even possible, in more complex environments, which could hinder the performance of the VQ-VAE-based methods.*

VQ-VAEs have been used to successfully compress and generate complex images in a number of works since the first paper on the architecture in 2017 ([van den Oord et al.](https://proceedings.neurips.cc/paper_files/paper/2017/file/7a98af17e63a0ac09ce2e96d03992fbc-Paper.pdf)). The authors of the original VQ-VAE paper even provide results to show how a VQ-VAE can reconstruct observations and predict believable, multi-step rollouts in DMLab specifically (Figures 5 & 7). The work since then has only gotten more impressive, with follow up work using VQ-VAEs to generate high-resolution images (e.g., [Razavi et al., 2019](https://proceedings.neurips.cc/paper/2019/file/5f8e2fa1718d1bbcadf1cd9c7a54fb8c-Paper.pdf), [Esser et al., 2021](https://arxiv.org/abs/2012.09841)) and videos (e.g., [Yan et al., 2021](https://arxiv.org/abs/2104.10157), [Walker et al., 2021](https://arxiv.org/abs/2103.01950)). The complexity of these datasets surpasses what is generally used in RL and gives us a high confidence that discretization works very well even in complex domains.
&nbsp;

&nbsp;

*2. Minigrid is inherently discrete unlike most real world problems, so these results may not generalize to more complex domains.*

As described in the “Why we used Minigrid” section, many of the experiments we performed, and the rigor of our experiments, were only possible due to our use of Minigrid. Running all of the same experiments in a more complex environment would not be possible (e.g. visualizing distributions over rollouts). After reading the reviews, our understanding is that the reviewers understand this and rather want to see a proof-of-concept to demonstrate that our findings are also applicable to more complex domains. If that is the case, then there are already several works that do exactly that on more complex domains ([Hafner et al., 2020](https://arxiv.org/abs/2010.02193), [Robine, 2020, Micheli et al., 2023](https://arxiv.org/abs/2010.05767), [Hafner et al., 2023](https://arxiv.org/abs/2301.04104)) and achieve strong results. Most importantly, [Hafner et al., (2020)](https://arxiv.org/abs/2010.02193) perform a direct comparison between continuous and discrete representations on over 50 Atari games (Figure 5) and find that discrete representations perform much better. Our setup is even similar to DreamerV2 in that both of our works use autoencoders to learn representations, and DreamerV2 uses a special case of a VQ-VAE (where the codebook is one-hot). We see our findings as complementary to these other works; they show that this is possible, and we help understand why that is the case and identify promising directions (continual learning).

One difference in evaluation is our focus on continual learning, whereas DreamerV2 looks at the episodic setting. Our experiments in Section 4, however, serve as evidence that the results in the episodic RL setting translate well into the continual RL setting (where the results are even better). We are currently working on an experiment in a more complex 3D environment, but we do not know if they will be done before the end of the discussion phase and are posting this now to give reviewers enough time to respond. We will post an update if we get the new results in time, but we believe that enough proof-of-concept experiments already exist in related work to justify our focus on depth and thoroughness by using a simpler environment.

---

### Author Response · Authors · 2023-11-23
**Thank you to the reviewers!**

We just uploaded a final draft of the manuscript that includes a new section on sparsity and some other minor changes made for clarity.

Thank you so much to all the reviewers that read through our work and took time out of their schedules to provide us with feedback. All of the reviewers had insightful comments that led to, what we think, is a better paper. We were also pleased to see several comments and questions that led us to think about interesting directions for future work.

Thank you for your time reviewing our work!

---

### Meta-Review · Area_Chair_vRz6 · 2023-12-06

**Metareview:**

(a) This paper carries out some systematic study on comparing the representation learning methods for RL and argue that discrete representations are indeed beneficial. Initially, the different representation learning methods, AE, FTA-AE, VQ-VAE, are compared for world model capability, then they showed the different effect carry over to the RL/Continual RL cases.

(b) The paper attempts systematically compare the benefit of discrete representation learning for RL setting.

(c) / (d) A simple Minigrid environment was used for validating their hypothesis. Multiple reviewers pointed out this is a major limitation for the work since it is not clear whether the findings of the paper could generalize to more complex settings. The authors have rebutted that the benefits of discrete representation learning have been already carried out in previous work and they tried to show why such benefit can be attained. Essentially, however, their assertion is the discrete representations do better job in world modeling under low resource constraints, but the meta-reviewer thinks systematically showing that assertion in more complex environment could be more convincing. Also, Continual RL experiments are on quite simple setting -- learning simple variations of minigrid --, and the conclusion seems also limited.

**Justification For Why Not Higher Score:**

As I mentioned above, as this is a paper with experimental validation, I believe the paper needs improvements to include additional extensive experiments on more complex environment to show that their claim -- discrete representations better model world model under resource constraints and is helpful for RL -- is indeed generalizable.

**Justification For Why Not Lower Score:**

N/A

---

### Decision · Program_Chairs · 2024-01-16

Reject